# Does Grappling Combat Sports Experience Influence Exercise Tolerance of Handgrip Muscles in the Severe-Intensity Domain?

**DOI:** 10.3390/sports12030066

**Published:** 2024-02-21

**Authors:** Rubens Correa Junior, Renan Vieira Barreto, Anderson Souza Oliveira, Camila Coelho Greco

**Affiliations:** 1Department of Physical Education, São Paulo State University, São Paulo 13506-900, Brazil; rubens.correa@unesp.br (R.C.J.); reenanvb@gmail.com (R.V.B.); 2Department of Materials and Production, Aalborg University, 9220 Aalborg, Denmark; oliveira@mp.aau.dk

**Keywords:** judo, jiu-jitsu, critical torque, neuromuscular fatigue

## Abstract

Successful performance in grappling combat sports (GCS) can be influenced by the fighter’s capacity to sustain high-intensity contractions of the handgrip muscles during combat. This study investigated the influence of GCS experience on the critical torque (CT), impulse above CT (W′), tolerance, and neuromuscular fatigue development during severe-intensity handgrip exercise by comparing fighters and untrained individuals. Eleven GCS fighters and twelve untrained individuals participated in three experimental sessions for handgrip muscles: (1) familiarization with the experimental procedures and strength assessment; (2) an all-out test to determine CT and W′; and (3) intermittent exercise performed in the severe-intensity domain (CT + 15%) until task failure. No significant differences were found in CT and neuromuscular fatigue between groups (*p* > 0.05). However, GCS fighters showed greater W′ (GCS fighters 2238.8 ± 581.2 N·m·s vs. untrained 1670.4 ± 680.6 N·m·s, *p* < 0.05) and exercise tolerance (GCS fighters 8.38 ± 2.93 min vs. untrained 5.36 ± 1.42 min, *p* < 0.05) than untrained individuals. These results suggest that long-term GCS sports training can promote increased tolerance to severe-intensity handgrip exercise and improved W′ without changes in CT or the magnitude of neuromuscular fatigue.

## 1. Introduction

Success in grappling-based combat sports (GCS) relies on one fighter controlling their opponent through long-lasting, strong handgrip actions [1]. Accordingly, GCS fighters’ physical and technical training often involves exercises to improve strength and tolerance of the hand and forearm muscles to high-intensity efforts [2].

The tolerance to high-intensity exercises is well-described by the critical power model, which suggests a hyperbolic relationship between the intensity of the exercise and the total duration that it can be sustained [3]. Two parameters can be derived from this relationship: the asymptote for the intensity measure (i.e., critical power, critical speed, or critical torque, CT) and the curvature constant (W prime, W′), which together allow the estimation of the tolerance to exercises performed above critical power [4,5]. Physiologically, the critical power/CT represents the highest rate at which oxidative metabolism stabilizes and is considered the boundary between the heavy and severe-intensity domains as well as a threshold for neuromuscular fatigue [6,7]. While W′ represents a finite amount of work that can be carried out above critical power [8].

Previous evidence demonstrates that the intensity associated with CT during handgrip exercise in untrained subjects is strongly influenced by oxygen availability, since when the oxygen supply is reduced, CT also decreases [9,10]. On the other hand, Bassan et al. (2019) [5] showed that when a short-term strength training program is performed, increases in both W′ and tolerance to severe-intensity exercise occur without changes in CT. Hence, the first hypothesis of the present study is that GCS fighters would have a greater W′ but a similar CT compared to the untrained individuals.

Neuromuscular fatigue can be defined as an exercise-induced reversible decline in muscle torque or power-generating capacity and usually results from impaired peripheral (i.e., sites distal or at the neuromuscular junction) and/or central (i.e., sites proximal to the neuromuscular junction) mechanisms of muscle contraction [11]. When exercise intensity exceeds the critical intensity, a progressive loss of metabolic homeostasis and neuromuscular efficiency rises to critical values, which results in an inability to sustain the work rate and, consequently, to continue the task [6,7]. Moreover, as exercise intensity is increased within the severe-intensity domain, the higher physiological requirements may accelerate the development of neuromuscular fatigue, which changes the rate of decline in maximal strength, contributes to a quick metabolic homeostasis perturbation, and reduces neuromuscular efficiency [12].

For instance, previous evidence demonstrated significant post-combat fatigue effects on upper-body endurance (~27%) and in the handgrip maximal strength of both hands (~8–12%) after simulated combats [13,14,15]. Hence, the second hypothesis of the present study is that GCS fighters would have a greater tolerance to severe-intensity exercise as well as greater neuromuscular fatigue at task failure compared to untrained individuals. Therefore, the purpose of this study was to investigate the influence of GCS experience on the CT, W′, tolerance, and neuromuscular fatigue development during severe-intensity handgrip contractions in GCS fighters and untrained individuals.

## 2. Materials and Methods

### 2.1. Participants

Eleven male recreational GCS fighters (30 ± 7 years, 81.2 ± 9.9 kg, 177 ± 4 cm) and twelve untrained male adults (24 ± 6 years, 79.5 ± 17.5 kg, 177 ± 5 cm) participated in the present study. The inclusion criteria for the fighters were two years of experience and at least three sessions a week, whereas untrained participants had not practiced any exercise within the past six months. The recreational fighters enrolled in this study reported 8 ± 5 years of GCS practice and a training frequency of 4 ± 1 sessions/week. Due to equipment limitations specially designed for left-handed individuals, only right-handed participants were selected for the study (the hand used for most daily tasks). Consequently, none of the participants had any previous musculoskeletal injuries on the dominant arm. All participants were instructed to arrive at the laboratory in a rested state for all sessions (i.e., avoid any strenuous activity in the previous 24 h). The participants were asked to avoid any food or drink containing caffeine for at least 3 h preceding the sessions. All participants signed an informed consent form approved by the institutional Ethics Review Board. The current study was conducted in accordance with the Declaration of Helsinki’s policy statement regarding the use of human participants and approved by the São Paulo State University’s Ethics Committee.

### 2.2. Experimental Procedures

All experimental sessions were conducted in a climate-controlled laboratory (21–23 °C) at the same time each day. Participants from both groups attended the laboratory on three different occasions, separated by at least 48 h. The first visit was designed to familiarize the participants with the experimental procedures, the acquisition of anthropometrical data (i.e., height, weight, and skin folds), the acquisition of ultrasonographic images of the forearm muscles, and the determination of electrostimulation current intensity. During the second visit, all participants performed a handgrip all-out test for the determination of CT and W′. In the third visit, the participants performed a handgrip test in the severe-intensity domain until task failure for the evaluation of neuromuscular fatigue and tolerance. Figure 1 shows an overview of the study design.

### 2.3. Dynamometer Settings

Participants were positioned in a sitting posture on the dynamometer seat with their shoulder joints flexed at 90° and their dominant arm supported by an arm attachment. The arm was aligned with the prehension with parallel grip attachment (Biodex Work-Simulation Tools, Shirley, NY, USA), maintaining a 0° abduction. This position ensured that the fixed lever was aligned with the proximal region of the second-to-fifth metacarpals and the movable lever was in the proximal region of the second-to-fifth phalanges of the hand. A bandage was applied to the hand to keep it in the same position throughout tests. Straps were used to secure participants’ hips and torsos fixed, ensuring the stability of involved joints in the tests, and avoiding strength production by other muscular groups.

### 2.4. Maximal Voluntary Contractions

MVC was performed to assess the handgrip maximal strength of the dominant arm. Warm-ups included 3 min arm-cycling in the upper-body ergometer (Cybex Metabolic System, Lumex, Inc., Ronkonkoma, NY, USA) and 10 submaximal isometric contractions (20% MVC) for all participants, followed by 3 min of rest. The MVC assessment involved three maximal isometric contractions (3 s contraction, 2 min rest) with the application of the twitch interpolation technique to evaluate the neuromuscular function (described below) [3]. Participants were informed to start the contraction as strong and fast as possible, receiving verbal encouragement to perform maximal effort for each contraction from the same investigator [3]. MVC corresponded to the peak torque attained during trials.

### 2.5. CT and W′ Determination

The CT and W′ of handgrip muscles were estimated using the methodology proposed by Burnley (2009). The 5 min all-out test consisted of 60 MVC (3 s effort, 2 s rest) [3]. Participants were informed about their MVC and instructed to reach or pass it during the first three contractions of the test [3]. Throughout the test, participants were strongly encouraged to perform a maximal effort at each contraction by the same investigator. Still, they were not informed about elapsed time or contractions to avoid any “pacing” strategy [3]. CT corresponded to the mean torque of the last six contractions, and W′ was determined as the sum of impulses above CT from each contraction using the area under the torque vs. time curve [3]. Custom-made scripts written on MatLab R2023a (MathWorks Inc., Natick, MA, USA) were used to analyze the data.

### 2.6. Exercise Tolerance

A constant-load isometric intermittent test (3 s contraction, 2 s rest) [12] was performed at an intensity corresponding to CT + 15% until exhaustion to evaluate exercise tolerance. Participants were informed to start the contraction quickly, hitting and keeping the target torque, resulting in a rectangular contraction [3,16]. Exhaustion corresponds to the first of three consecutive contractions where participants could not reach the target torque despite verbal encouragement [3,12]. MVCs were performed before the test and immediately after exhaustion to quantify neuromuscular fatigue (Figure 2).

### 2.7. Neuromuscular Fatigue Assessment

Transcutaneous electrically evoked contractions were induced by a high-voltage constant-current stimulator (DS7AH, Digitimer, Welwyn Garden City, UK), delivering a square-wave 200-µs stimulus to the median nerve on the anterior antebrachial region of the arm with a maximal voltage of 400 V. Stimulation electrodes were attached to the dominant arm’s antebrachial region for median nerve stimulation. The cathode (0.5 cm diameter, Dermatrode, American Imex, Irvine, CA, USA) placement was over the median nerve on the anterior antebrachial region, with the anode (0.5 cm diameter, Dermatrode, American Imex, Irvine, CA, USA) placed proximal to the olecranon on the posterior brachial region of the arm [17,18]. Global fatigue was determined by the difference between MVCs performed before and immediately after the exercise tolerance test during the third session [12]. An incremental amperage electrostimulation protocol was performed during the first session to determine maximal stimulation of the median nerve. Starting with 100-mA stimulus intensity and increasing by 25 mA until evoked torque reaches a plateau, ensuring that supramaximal stimulation was applied, 30% of the corresponding plateau value was increased [18]. For assessing neuromuscular function, the twitch interpolation technique was used, delivering supramaximal stimulation into the MVC plateau to get a superimposed twitch and 1.5 s after contraction to get a resting twitch [19]. Decreases in resting twitch and voluntary activation of MVCs performed before and immediately after exercise tolerance tests determine peripheral and central fatigue, respectively [19,20]. Voluntary activation was estimated using the following equations: (1) when superimposed twitch occurs at MVC [3] and (2) corrected for when superimposed twitch did not occur at MVC [17,18].
%Voluntary activation = [1 − (superimposed twitch/resting twitch)] ∙ 100(1)
%Voluntary activation = [1 − (force before superimposed twitch/MVC) ∙ (superimposed twitch/resting twitch)] ∙ 100(2)
where MVC is a maximal voluntary contraction.

### 2.8. Electromyography

Electromyography was acquired using bipolar Ag–AgCl surface electrodes. For their placement, the skin was shaved, lightly abraded, and cleaned with alcohol. Electrodes were placed on the flexor digitorum superficialis muscle [21] with a center-to-center distance of 2 cm apart. The reference electrode was placed on the patella bone. The starting signal was kept at less than 5 μV. Electromyography data were acquired using the biological signal acquisition model Miotool (Miotec, Porto Alegre, Brazil), amplified by 10× (i.e., forming with a pre-amplifier, a total gain of 1000×). Converting analog signals to digital was performed by an -A/D board with an input range of −5 to +5 Volts. Miograph software (Miotec, Porto Alegre, Brazil) acquired electromyography signals with a sampling frequency calibrated at 2000 Hz. Electromyography data were band-pass filtered (20–500 Hz) using a digital fourth-order zero-lag Butterworth filter [22], corresponding over a 1 s period (i.e., 0.5 s before and after the peak torque) from each muscle contraction assessed using custom-made scripts written on MatLab^®^ 7.0 (MathWorks Inc., Natick, MA, USA). The electromyography muscle fatigue index was assessed using signals derived from the amplitude (root mean square, RMS) and frequency (median frequency, MDF) domains during the third session [23]. RMS data from each submaximal contraction throughout the exercise tolerance test were normalized to their peak value of pre-exercise MVC. Fatigue-related changes in muscle activation throughout the exercise tolerance test were assessed by neuromuscular efficiency analysis (i.e., torque/RMS) as well as changes in RMS and MDF between the first and last 30 s.

### 2.9. Ultrasonography

Muscle thickness was acquired using a portable ultrasound device (ProSound 2, ALOKA, Tokyo, Japan) in B-mode function with a 38 mm probe and a capture frequency of 9.0 MHz during the first session. Flexor digitorum superficialis (humeroulnar head) was measured transversally on the lateral anterior portion of the forearm (Figure 3A), 30% proximal to the styloid process and the olecranon [24]. Positions were referenced and marked with semi-permanent ink. The probe was aligned with traversal ink markers on these muscles and coated with a water-soluble transmission gel to give acoustic contact between the skin and the transducer. The same investigator acquired and analyzed images cautiously, avoiding compression of the dermal surface. The muscle thickness was assessed using ImageJ 1.42q software (National Institutes of Health, Bethesda, MD, USA), calibrated with a fixed distance scale of ultrasound images. Muscle thickness was considered the distance between the higher superficial-to-deep fasciae of the muscle (Figure 3B).

### 2.10. Torque Data Acquisition and Analysis

Torque data are acquired using an isokinetic dynamometer (Biodex System 3, Shirley, NY, USA) synchronized with a biological signal acquisition model (Miotool, Miotec, Porto Alegre/RS, Brazil), calibrated according to the manufacturer’s instructions. The data were sampled at 2000 Hz and analyzed using custom-made scripts written on MATLAB (MathWorks Inc., Natick, MA, USA). A digital fourth-order zero-lag Butterworth filter smoothed torque curves with a cutoff frequency of 20 Hz [22].

### 2.11. Statistical Analyses

Data are presented as mean ± standard deviation (SD). Normality and homogeneity of the data were confirmed by the Shapiro-Wilk and Levene tests, respectively. Data comparisons between groups were performed using the Student’s *t*-test (when samples met parametric conditions) or the Mann–Whitney U test (when samples did not meet parametric conditions). The variance of the MVC, resting twitch, voluntary activation, RMS (% MVC), and MDF during the exercise tolerance test were assessed using two-way repeated-measures ANOVAs (time vs. group). Additionally, the magnitude of the difference between comparisons was calculated using Cohen’s d effect size (ES) and partial eta square (η_p_^2^). Interpretation of the magnitude of difference followed the criteria indicated by Cohen (1998) for the *t*-test (ES) as trivial (<0.2), small (0.2–0.4), medium (0.4–0.8), and large (>0.8) [25]. While for ANOVA (η_p_^2^) as small (0.01≤ η_p_^2^ < 0.06), medium (0.06≤ η_p_^2^ < 0.14), and large (≥0.14). Pearson’s correlation coefficient was used to assess associations between variables [25]. Pearson’s correlation results were interpreted following the criteria indicated by Schober (2018) as negligible (0.00–0.10), weak (0.10–0.39), moderate (0.40–0.69), strong (0.70–0.89), and very strong (0.90–1.00) correlations [26]. The significance level was set at *p* < 0.05. Data were analyzed using a statistical software package (SPSS Version 20.0; IBM, New York, NY, USA).

## 3. Results

### 3.1. Anthropometrical Data

No significant differences between groups were found for any anthropometrical data (age, height, body mass, lean body mass, and fat percentage). GCS fighters presented greater muscle thickness (fighters 20.8 ± 2.15 mm vs. untrained 17.3 ± 2.30 mm, *p* < 0.01; ES = 1.61, large) than untrained participants (Figure 3C). Pearson’s correlation analysis showed a moderate and significant correlation between muscle thickness with W′ (r = 0.521, *p* < 0.01) and exercise tolerance (r = 0.515, *p* < 0.01) within both groups.

### 3.2. Neuromuscular Function

No significant differences between groups were found for baseline MVC (fighters 79.9 ± 13.2 N·m vs. untrained 78.3 ± 19.1 N·m, *p* > 0.05; ES = 0.10, trivial), MVC relative to weight (fighters 1.00 ± 0.23 N·m/kg vs. untrained 1.02 ± 0.34 N·m/kg, *p* > 0.05), and voluntary activation (fighters 96.4 ± 3.41% vs. untrained 95.6 ± 4.04%, *p* > 0.05; ES = 0.20, trivial).

### 3.3. CT, W′, and Exercise Tolerance

No significant differences between groups were found for CT (*p* > 0.05; ES = 0.22, small) and CT (% MVC) (*p* > 0.05). However, Student’s *t*-test analysis showed that fighters had greater W′ than untrained (*p* < 0.05; ES = 0.89, large) (Table 1). Figure 4 shows the mean torque from each contraction during an all-out test between groups. Similarly, for exercise tolerance, fighters had longer exercise tolerance than untrained (fighters 8.38 ± 2.93 min vs. untrained 5.36 ± 1.42 min, P < 0.05; ES = 1.33, large). Estimated W′ from the all-out test and accumulated W′ during severe-intensity contractions were similar (*p* > 0.05) for both fighters (all-out 2238.8 ± 581.2 N·m·s vs. severe exercise 1985.6 ± 413.1 N·m·s, ES = 0.50, medium) and untrained (all-out 1670.4 ± 680.6 N·m·s vs. severe exercise 1385.8 ± 382.5 N·m·s, ES = 0.52, medium). Pearson’s correlation analysis showed a strong and significant correlation between estimated W′ from all-out and accumulated W′ during severe-intensity contractions for fighters (r = 0.749, *p* < 0.05). However, a moderate but non-significant correlation was found for the untrained (r = 0.417, *p* > 0.05). Similarly, a moderate and significant correlation was found between accumulated W′ during severe-intensity contractions and exercise tolerance for fighters (r = 0.675, *p* < 0.05). However, a weak and non-significant correlation was found for the untrained (r = 0.197, *p* > 0.05).

### 3.4. Neuromuscular Fatigue during Handgrip Severe-Intensity Contractions

No significant differences were found for pre-exercise MVC (*p* > 0.05; ES = 0.28, small), resting twitch (*p* > 0.05; ES = 0.33, small), and voluntary activation (*p* > 0.05; ES = 0.23, small) (Table 2). Two-way repeated measures ANOVA showed a significant time-effect for MVC (F = 162.2; *p* < 0.05; η_p_^2^ = 0.89, large), resting twitch (F = 57.5; *p* < 0.05; η_p_^2^ = 0.73, large), and voluntary activation (F = 52.3; *p* < 0.05; η_p_^2^ = 0.71, large) for both groups. No significant differences between groups and interaction effects for any variable were found (*p* > 0.05). Relative pre-to-post comparisons showed no significant differences in any variable at exhaustion between groups (Table 2).

### 3.5. Electromyography

Two-way repeated measures ANOVA showed a significant time effect for RMS (F = 18.5; *p* < 0.05; η_p_^2^ = 0.47, large) and MDF (F = 43.3; *p* < 0.05; η_p_^2^ = 0.67, large) for both groups. However, no significant differences between groups and interaction effects were found (*p* > 0.05). Relative pre-to-post comparisons showed no significant differences for RMS (fighters 42.9 ± 29.6% vs. untrained 60.8 ± 68.6%, *p* > 0.05) and MDF (fighters −10.4 ± 10.2% vs. untrained −15.0 ± 8.88%, *p* > 0.05) between groups. Also, relative pre-to-post comparisons based on neuromuscular efficiency analysis showed no significant differences between groups (fighters −30.2 ± 14.9% vs. untrained 32.6 ± 26.0%, *p* > 0.05). Figure 5 shows an overview of the effect of severe-intensity contractions on the first and last 30 s of electromyography signals.

## 4. Discussion

The main purpose of this study was to investigate the influence of GCS training experience on tolerance to severe-intensity handgrip exercise by comparing trained and untrained participants. Secondly, it was investigated whether GCS training modulates CT, W′, and the mechanisms of neuromuscular fatigue. It was hypothesized that fighters would have greater tolerance, W′, and neuromuscular fatigue than untrained. The main findings were that fighters had greater W′ (~34%) and tolerance to severe-intensity exercise (~56%), with similar CT compared to the untrained. Thus, the hypotheses of this study were accepted, showing that GCS training can improve tolerance to severe-intensity handgrip exercise by improving the capacity to accumulate greater amounts of W′.

In this study, fighters had a similar neuromuscular function (i.e., MVC and voluntary activation) compared to the untrained. It could be related to the training status of fighters (recreational level) from this study, which includes fighters with low training volume in GCS (e.g., three to five sessions/week with sessions lasting ~75 min). In contrast, international and national GCS athletes commonly train ~3 times a day, including GCS training modalities (e.g., technical, specific exercises, and sparring) and strength/conditioning sessions [27]. Supporting this notion, most fighters (8 of 11) in this study were not engaged in systematic strength training, especially aimed at maximal strength development. Therefore, GCS fighters’ training status might be a factor in explaining the similarity of baseline neuromuscular function, since previous studies have shown increases in MVC (~17–19%) when a systematic strength training program is adopted [16,28].

Although MVC did not significantly differ between groups, tolerance to handgrip severe-intensity exercise was greater (~56%) for fighters than untrained. Suggesting that the time course of long-term GCS adaptations on fighters’ physical fitness should depend on the specificity of the task [29], muscle endurance first improved, probably requiring higher volumes of maximal strength training to improve their MVC. Similar to a previous longitudinal study that showed an increase (~26%) in W′ without CT changes after a strength training program [16], fighters (i.e., trained individuals) showed greater (~34%) W′ than untrained, with similar CT between groups (Table 1). Suggesting that adaptations induced by GCS training were more comparable to those induced by high-load than low-load strength training because high-load training was shown to increase work efficiency and endurance [28,30]. While low-load training increases oxidative parameters [31].

Similarly, it is well known that high-load strength training induces muscle protein synthesis activation, which mediates muscle hypertrophy [29,32]. Fighters had greater (~20%) muscle thickness of flexor digitorum superficialis muscle than untrained, which indicates that along with the results, fighters’ handgrip muscles were more hypertrophied compared to the untrained. Nevertheless, together with the fighters’ training status, the level closer to maximum voluntary activation (~96%) shown by both groups could be a factor in explaining the MVC similarities between groups because probably all the motor units are recruited under these conditions [20]. Indeed, multiple factors could influence muscle contractions, and it is possible to state, based on the findings of this study, that adaptations induced by any training are specificity-dependent [29].

Factors related to oxidative metabolism have a crucial role in determining the maximum intensity at which a steady state of physiological responses can be achieved (i.e., CT) [6,12]. Additionally, oxygen delivery, fiber type composition (type I), muscle capillarity, and aerobic stimulus resulting from continuous or intermittent training [33,34,35,36] have all been identified as contributors to an elevated CT. Consistent with these statements, CT did not significantly differ between groups, which could be related to a low stimulus for continuous low-intensity contractions during combat. Although, during combat, handgrip muscles perform long-lasting strong contractions [37], it might suggest that for our fighters’ group, GCS training probably did not induce improvements in oxidative components. Supporting this notion, previous studies revealed that GCS fighters do not have highly developed aerobic capacity [1,38]. Improving aerobic function (i.e., aerobic capacity and power) might be a relevant factor in fighters’ performance because these adaptations promote increased resistance to fatigue and a faster recovery process [32].

In line with previous studies [39,40], it was shown in the current study that there were moderate and significant correlations between muscle thickness and W′ (r = 0.521) and exercise tolerance (r = 0.515). The underlying mechanisms related to the capacity to accumulate W′ and tolerate exercises performed in the severe-intensity domain are multifactorial [41]. For instance, it relies on several factors, including glycogen and PCr content [42,43], oxygen delivery [17,18], and the synchronization and recruitment of motor units [33]. Since analyses related to the loss of neuromuscular function during exercise (i.e., neuromuscular efficiency analysis) and pre-to-post decrease (i.e., neuromuscular fatigue analysis) showed no differences between groups, it indicates that the greater tolerance of GCS fighters compared to untrained individuals can be explained by their capacity to accumulate greater amounts of W′, which directly contributes to fighters tolerating more contractions until task failure.

Supporting the concept that W′ is a finite amount and their size dictates the tolerable duration during severe-intensity exercises, W′ did not significantly differ between the values estimated from the all-out test and accumulated during severe-intensity exercise for both groups. However, the moderate ES could be related to the distinct bioenergetic characteristics between an all-out and a constant-load test [44]. Moderate and significant correlations (r = 0.675) were found between accumulated W′ during severe-intensity exercise and exercise tolerance for fighters, while weak and non-significant correlations (r = 0.197) were found for the untrained. Similarly, a significant correlation (r = 0.749) was found between estimated W′ from all-out and accumulated W′ during severe-intensity exercise for fighters, while a non-significant correlation (r = 0.417) was found for the untrained. These findings underscore the W′ dynamics between different testing protocols, particularly when considering two distinct populations with different training experiences. Also, the results from the correlation analyses among trained individuals between W′ of different testing protocols and exercise tolerance reveal a notable agreement, which indicates a positive association among those with a higher W′. Nevertheless, the results for the untrained may be related to their relative inexperience with high-intensity contractions and exercises until task failure, especially all-out tests. The lack of significant correlations could suggest that the lowest exposure to such physiological demands might impact the relationship between W′ of different testing protocols and exercise tolerance in this population.

Neuromuscular function assessment before the test to exhaustion did not significantly differ between groups, which includes similar results to those performed during the second session (Figure 1). ANOVAs with η_p_^2^ analysis revealed a large and significant time effect for any variable within both groups, which indicates a decline in MVC (~35–72%), voluntary activation (~5–29%), and resting twitch (~13–50%) induced by severe-intensity exercise for both groups. Our findings were comparable to previous studies assessing neuromuscular fatigue development after severe-intensity contractions of handgrip muscles [17,18]. Also, relative pre-to-post analysis showed no significant differences in MVC, voluntary activation, and resting twitch between groups (Table 2). ANOVAs with η_p_^2^ analysis of electromyography signals revealed a large and significant time effect for RMS and MDF, where RMS significantly increased and MDF decreased for both groups (Figure 5). However, ANOVAs, neuromuscular efficiency analysis, and relative pre-to-post behaviors in the RMS and MDF indicate no significant differences between groups. Together, these comprehensive findings suggest that while severe-intensity exercise-induced consistent neuromuscular changes in both groups, the groups did not differ between them in their responses to the loss of neuromuscular function. Thus, our findings demonstrated that repeated handgrip contractions performed within the severe-intensity domain lead to significant reductions in the whole neuromuscular system.

### Limitations

Some limitations should be considered when interpreting the present results. Analyses during fights were not performed in this study, which limits the direct association between handgrip performance during tests and combats. Future research is needed to investigate the associations of CT, W′, and neuromuscular fatigue mechanisms with GCS training sessions and combat performance. Moreover, further investigations involving different GCS weight classes, training status (i.e., not only recreational level GCS fighters), and genders may contribute to a better understanding of the influence of GCS practice on tolerance and fatigue during severe-intensity handgrip exercise.

## 5. Conclusions

The findings of the current study showed that fighters’ greater tolerance during handgrip severe-intensity contractions can be related to their greater ability to accumulate greater amounts of W′ compared to untrained individuals. These findings may guide scientists, coaches, and fighters to understand the applicability of CT and W′ in the capacity to sustain high-intensity contractions of handgrip muscles for longer periods, incorporating specific training loads prescribed based on intensity domains to promote a CT and W′ improvement.

## Figures and Tables

**Figure 1 sports-12-00066-f001:**
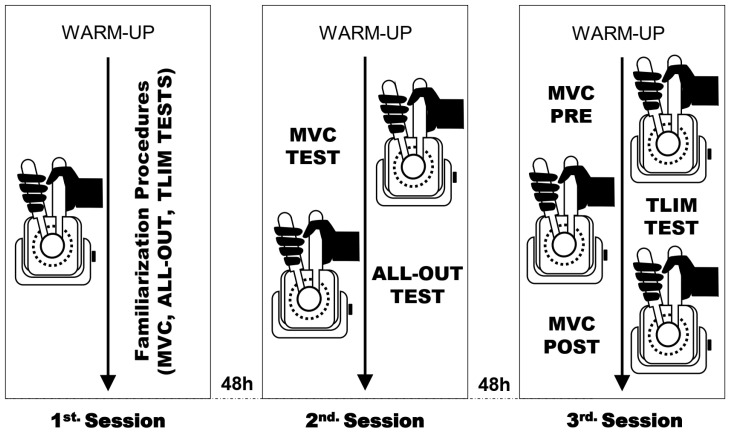
Study design. Abbreviations: MVC, maximal voluntary contraction; TLIM, exercise tolerance test.

**Figure 2 sports-12-00066-f002:**
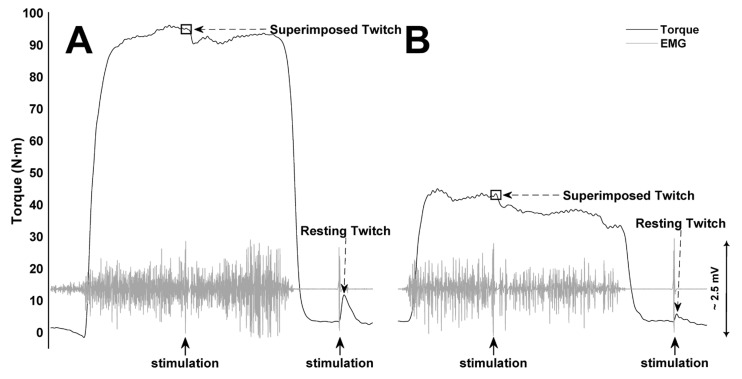
Handgrip maximal voluntary contractions. Torque output from pre- (**A**) and post-exercise maximal voluntary contractions (**B**) from a representative participant.

**Figure 3 sports-12-00066-f003:**
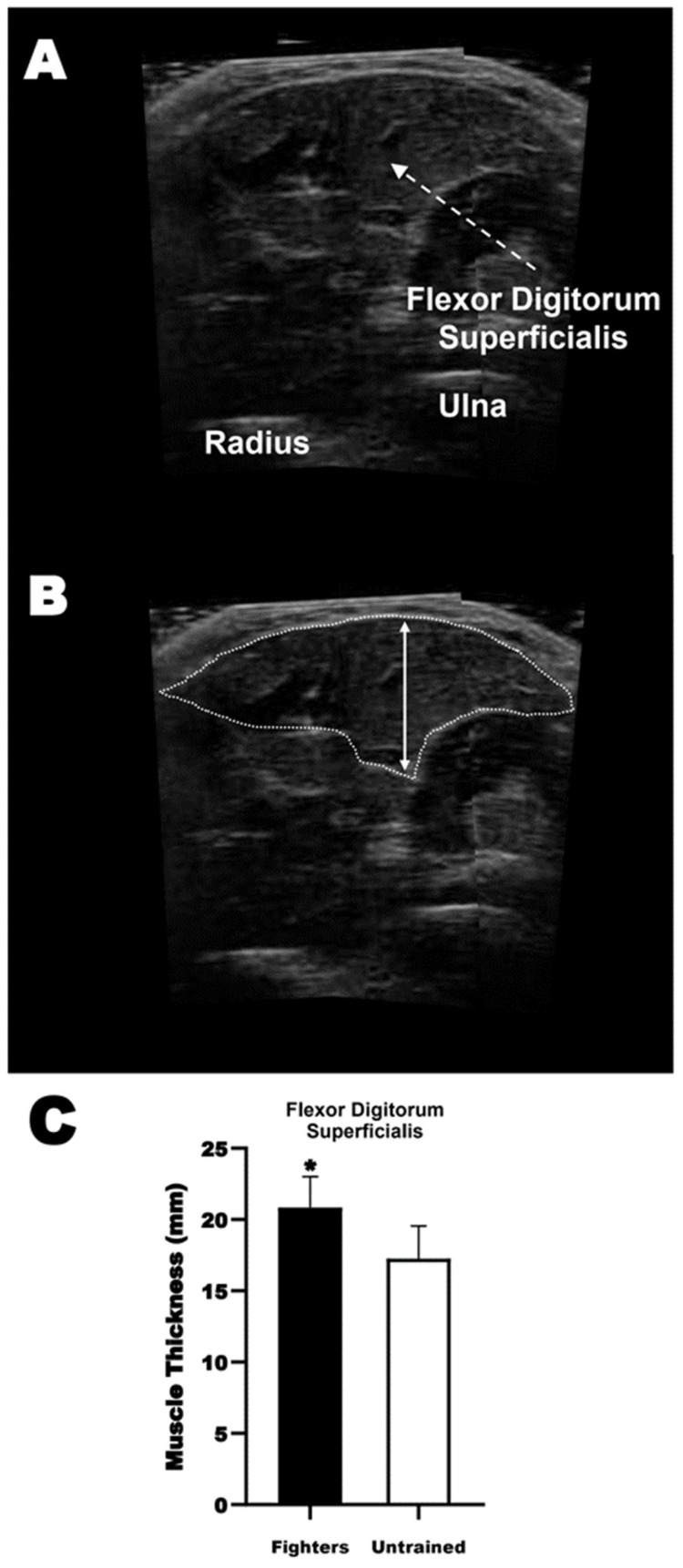
Muscle thickness. (**A**) shows the ultrasound assessment of flexor digitorum superficialis (humeroulnar head); (**B**) shows the methodological approach to determine muscle thickness; and (**C**) shows the results as mean ± SD of muscle thickness for both groups. This Symbol (*) means that GCS fighters had a significant grater flexor digitorum muscle thickness compares to the untrained individuals. * *p* < 0.05 compared to the untrained individuals.

**Figure 4 sports-12-00066-f004:**
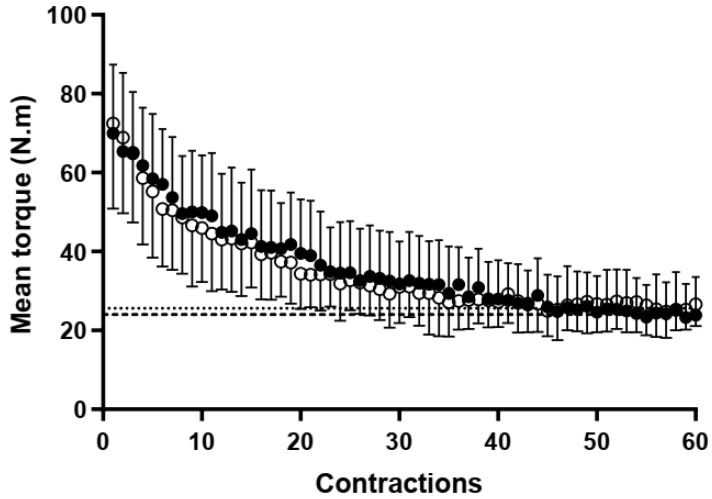
5 min handgrip all-out test. Mean torque ± SD during 5 min all-out test from GCS fighters (●) and untrained participants (○). The dashed and dotted lines correspond to the CT of fighters and untrained, respectively.

**Figure 5 sports-12-00066-f005:**
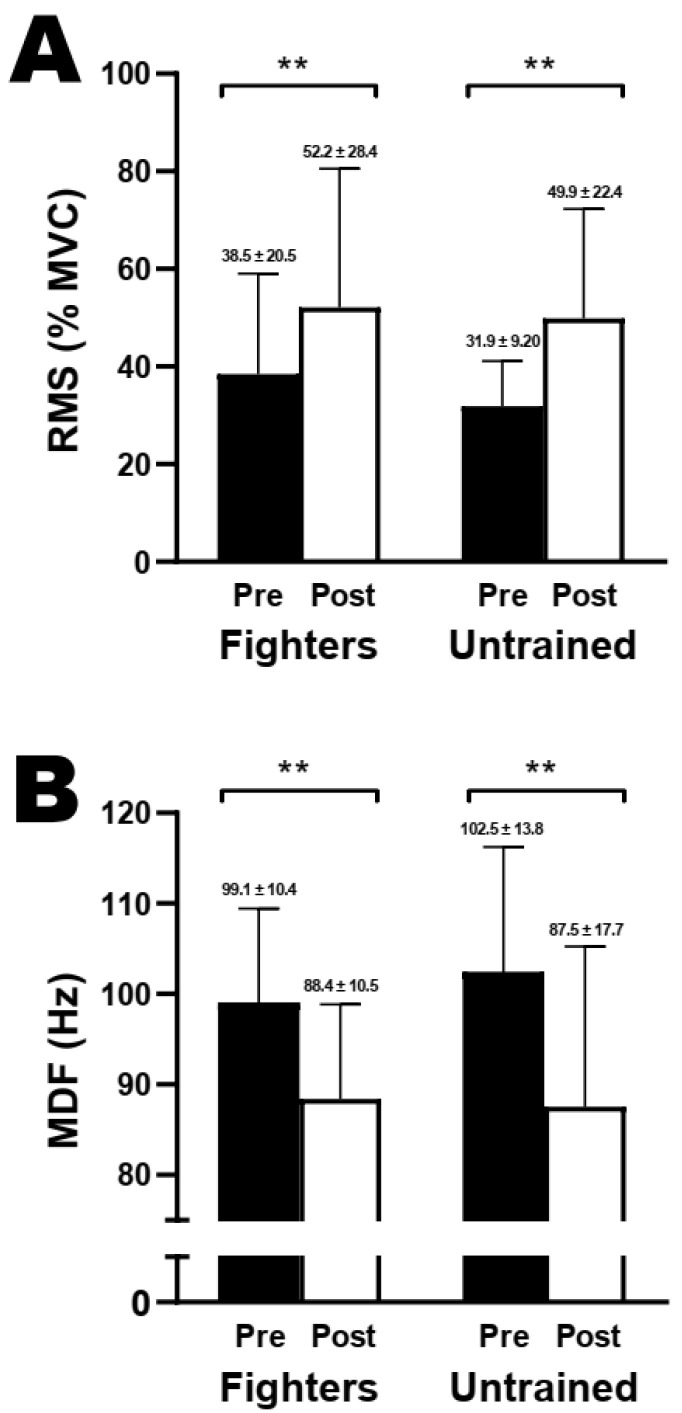
Fatigue-related changes in the RMS and MDF during handgrip severe-intensity contractions. Mean ± SD from median frequency (MDF), (**A**), and root mean square (RMS), (**B**) of the first and last 30 s during severe-intensity contractions until exhaustion. Abbreviations: MVC, maximal voluntary contraction; GCS, grappling combat sports. ** *p* < 0.05 compared with pre-exercise.

**Table 1 sports-12-00066-t001:** Critical torque and impulse above critical torque.

	Fighters (*n* = 11)	Untrained (*n* = 12)
CT (N·m)	24.0 ± 8.49	25.6 ± 5.84
CT (%MVC)	29.9 ± 8.67	33.5 ± 7.60
W′ (N·m·s)	2238.8 ± 581.2	1670.4 ± 680.6 *

Mean ± SD values of CT, critical torque; W′, impulse above critical torque. Abbreviations: MVC, maximal voluntary contraction. * *p* < 0.05 compared to the fighters.

**Table 2 sports-12-00066-t002:** Neuromuscular function before (pre) and after (post) severe-intensity contractions until exhaustion.

	Fighters (*n* = 11)			Untrained (*n* = 12)			
	Pre	Post	% Changes in Task Failure	Pre	Post	% Changes in Task Failure	*p* Value
MVC (N·m)	83.2 ± 19.2	35.4 ± 9.45 **	−56.4 ± 11.2	77.8 ± 19.5	40.4 ± 9.23 **	−47.3 ± 9.36	0.065
Voluntary Activation (%)	96.5 ± 3.51	84.9 ± 6.96 **	−11.9 ± 7.76	95.6 ± 4,66	87.9 ± 6.71 **	−7.97 ± 5.11	0.140
Resting Twitch (N·m)	11.3 ± 2.32	7.78 ± 2.10 **	−30.3 ± 18.1	12.0 ± 1.96	8.52 ± 1.60 **	−28.0 ± 14.3	0.806

Mean ± SD values of maximal voluntary contraction (MVC), voluntary activation, and resting twitch. ** *p* < 0.05 compared with pre-exercise.

## Data Availability

Data will be made available for readers upon reasonable request.

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
