# Peer review of "Does Grappling Combat Sports Experience Influence Exercise Tolerance of Handgrip Muscles in the Severe-Intensity Domain?"

_sports, 2024, doi:10.3390/sports12030066_

Round 1

Reviewer 1 Report

Comments and Suggestions for Authors

The article was written very carefully. In my opinion, it will be very useful for practitioners-trainers involved in  in sports and martial arts. It would be repeating the research  on larger sample of people-athletes.

Reviewer 2 Report

Comments and Suggestions for Authors

Thank you for the opportunity to review these interesting paper. I would like to make a few comments in advance: The chosen approach and the questions (hypotheses) are comprehensible from the authors' point of view and the methodological and statistical evaluation is convincing. The description of the accuracy of the test implementation, the test exercises and the various methods should be emphasized positively. The presentation of the results is comprehensible at all times and corresponds to the usual procedure. 

I see major problems in the following points:

Is the research design generally suitable to answer the hypotheses? Is it not the intention of performance-specific training that specific adaptations should be achieved? It is all the more surprising that no significant differences were found between the competitive athletes and the untrained in numerous variables. This means that the results are predetermined by the selection of the sample. How would the results have turned out if climbers had been selected as the control group? Especially with regard to grip strength. I therefore consider the selected sample to be unsuitable for answering the question. Shouldn't competitive athletes at different performance levels, different training durations and competition levels be examined to answer the questions? Comparing apples with pears always leads to differences (all the more surprising that only a few significant differences were found between the two groups). I therefore consider the study's approach to be problematic and of little use.  Since the two groups were selected "arbitrarily" (no randomization, stratification, statistical twin, etc.), I see a major problem with this.

Reviewer 3 Report

Comments and Suggestions for Authors

In the present study, various aspects of handgrip strength were investigated between two groups: grappling combat sports fighters vs untrained individuals. These differences were assessed through well-studied characteristics in the literature: critical torque (CT), impulse above CT (W’), tolerance, and neuromuscular fatigue. According to the authors, the identified differences between the groups, as evidenced, may be attributed to experience and continuous practice in sports that employ this pattern of manual force. This research holds significance for advancing the field of study regarding specific adaptations in sports and their reversibility. Especially noteworthy is the absence of differences in the CT variable, suggesting that the dynamics of endurance adaptations may be more easily observable than those related to strength.

The study's design is sound. The two hypotheses are correctly stated, and the objective is rational. The authors demonstrated meticulous attention to detail in the methods and results sections. The discussion is valuable and well-structured. Although the overall paper is robust, this reviewer would like to highlight some issues that need to be adjusted:

1. In the abstract, the CGS abbreviation is not defined.

2. In section 2, use ‘Material and Methods’ instead of ‘Material e Methods’.

3. In section 2.1, the recreational level of participants in the CGS group should be declared within this section, rather than in the Discussion section, as the authors did.

4. In section 2.4, the authors assessed the participants' dominant arm (right arm). It is essential to clarify whether all participants were right-handed and provide details on how the authors assessed dominance. This information should be declared in section 2.1.

5. What does the expression ‘by the same investigator’ mean? This expression occurs several times.

6. In section 2.6, a Ref is missed

7. In section 2.7, fix ‘femoral and median nerves’

8. In section 2.8, RMS and MDF abbreviations are not defined. Moreover, the authors should briefly explain these two variables for a reader-friendly presentation.

9. In section 2.11, the authors must state which variables did not verify normality. Moreover, I did not find any U (Mann-Whitney) results reported.

10. Figure captions should be consolidated into a single paragraph; avoid splitting the caption. Also, ensure that any abbreviations used in each figure are defined in the caption.

11. This reviewer encourages the authors to improve all figures by using normalized font sizes for all text within the figures (see Fig 5). In addition, to make better use of sheet width, it is suggested that the organization and size of the figures be optimized.

12. Regarding the article title, this reviewer believes that the term ‘muscles’ may not be appropriate since the study only evaluates one muscle (FDS).

Comments on the Quality of English Language

Minor issues

Round 2

Reviewer 2 Report

Comments and Suggestions for Authors

The revisions were target-oriented and the helpful!

Comments on the Quality of English Language

The revision was manageable but effective.